# A loss-of-function mutation in *RORB* disrupts saltatorial locomotion in rabbits

Miguel Carneiro[1,2]☯*, Jennifer Vieillard[3]☯, Pedro Andrade[1], Samuel Boucher[4], Sandra Afonso[1], José A. Blanco-Aguiar[1], Nuno Santos[1], João Branco[2], Pedro J. Esteves[1,2], Nuno Ferrand[1,2,5], Klas Kullander[3], Leif Andersson[6,7,8]*

**1** CIBIO/InBIO, Centro de Investigação em Biodiversidade e Recursos Genéticos, Universidade do Porto, Vairão, Portugal, **2** Departamento de Biologia, Faculdade de Ciências, Universidade do Porto, Porto, Portugal, **3** Department of Neuroscience, Uppsala University, Uppsala, Sweden, **4** Labovet Conseil (Réseau Cristal), Les Herbiers Cedex, France, **5** Department of Zoology, Faculty of Sciences, University of Johannesburg, Auckland, South Africa, **6** Science for Life Laboratory Uppsala, Department of Medical Biochemistry and Microbiology, Uppsala University, Uppsala, Sweden, **7** Department of Veterinary Integrative Biosciences, College of Veterinary Medicine and Biomedical Sciences, Texas A&M University, College Station, Texas, United States of America, **8** Department of Animal Breeding and Genetics, Swedish University of Agricultural Sciences, Uppsala, Sweden

☯ These authors contributed equally to this work.
* miguel.carneiro@cibio.up.pt (MC); leif.andersson@imbim.uu.se (LA)

**Data Availability Statement:** The data underlying the results presented in the study are available in the Sequence Read Archive (www.ncbi.nlm.nih. gov/sra) under the bioproject PRJNA559371.

## Abstract

Saltatorial locomotion is a type of hopping gait that in mammals can be found in rabbits, hares, kangaroos, and some species of rodents. The molecular mechanisms that control and fine-tune the formation of this type of gait are unknown. Here, we take advantage of one strain of domesticated rabbits, the *sauteur d'Alfort*, that exhibits an abnormal locomotion behavior defined by the loss of the typical jumping that characterizes wild-type rabbits. Strikingly, individuals from this strain frequently adopt a bipedal gait using their front legs. Using a combination of experimental crosses and whole genome sequencing, we show that a single locus containing the RAR related orphan receptor B gene (*RORB*) explains the atypical gait of these rabbits. We found that a splice-site mutation in an evolutionary conserved site of *RORB* results in several aberrant transcript isoforms incorporating intronic sequence. This mutation leads to a drastic reduction of RORB-positive neurons in the spinal cord, as well as defects in differentiation of populations of spinal cord interneurons. Our results show that *RORB* function is required for the performance of saltatorial locomotion in rabbits.

## Author summary

Rabbits and hares have a characteristic jumping gait composed of an alternate rhythmical movement of the forelimbs and a synchronous bilateral movement of the hindlimbs. We have now characterized a recessive mutation present in a specific strain of domestic rabbits (*sauteur d'Alfort*) that disrupts the jumping gait. The mutation causing this defect in locomotion pattern occurs in the gene coding for the transcription factor RORB that is normally expressed in many regions of the nervous system especially in the spinal cord dorsal horn. Our results show that expression of RORB is drastically reduced in the spinal

**Funding:** This work was supported by the Fundação para a Ciência e Tecnologia (FCT, https://www.fct.pt) through POPH-QREN funds from the European Social Fund and Portuguese MCTES (CEECINST/00014/2018/CP1512/CT0002 and IF/00283/2014/CP1256/CT0012); by FEDER funds through the COMPETE program and Portuguese national funds through FCT (projects PTDC/CVT/122943/2010 and PTDC/BIA-EVL/30628/2017); by the project NORTE-01-0145-FEDER-AGRIGEN, supported by the Norte Portugal Regional Operational Programme (NORTE2020) under the PORTUGAL 2020 Partnership Agreement and through the European Regional Development Fund (ERDF); by grants from the Swedish Research Council (KK, LA), the Knut and Alice Wallenberg Foundation (LA), the Swedish Brain Foundation (KK) and the Swedish Foundation for Cooperation in Research and Higher Education (KK); and by travel grants to M.C. (COST Action TD1101). J.V. was supported by a postdoctoral contract from Stiftelsen Promobilia. The funders had no role in study design, data collection and analysis, decision to publish, or preparation of the manuscript.

**Competing interests:** The authors have declared that no competing interests exist.

cord of affected rabbits which results in a developmental defect. This study is an advance in our understanding how locomotion is controlled in vertebrates.

## Introduction

The development of coordinated limbed locomotion is an important life-history trait that is key for individual survival and reproduction. The interlimb coordination pattern used by animals during locomotion is called gait, which results from an accurate integration of sensory information and motor commands [1,2]. This integration is responsible for determining rhythm, flexor–extensor muscle activity within a limb, and left–right limb coordination, and is largely controlled by central pattern generator (CPG) neural networks located within the spinal cord [3–6]. Most mammals share the capability of switching between different gaits with speed or terrain (e.g. walking, trotting, galloping). Gait pattern also differs considerably between species, ranging from bipedal to quadrupedal, and from left-right alternation observed in most mammals to left-right synchronous that allows hopping gaits in kangaroos, most lagomorphs, and some rodents [7]. Despite intense interest in the biomechanical, morphological, and physiological adaptations that characterize distinct types of locomotion in vertebrates [8], the genetic, molecular, and developmental bases underlying differences between individuals and species have seldom been reported [6,9–12].

Among mammals, rabbits and hares have a particular saltatorial locomotion pattern characterized by different flexion and extension patterns of the forelimbs (alternate rhythmical) and hindlimbs (synchronous bilateral), with a relatively more pronounced extension than flexion in the hindlimbs, leading to the characteristic jumping [13]. One strain of domesticated rabbits, the *sauteur d'Alfort* (hereafter referred to only as *sauteur*), is known to differ from this pattern and exhibits an abnormal locomotion behavior (Fig 1A) [14–19]. At slow speed, during the swing phase, they lift excessively their hindlimbs. At higher speed, the movements of the hindlimbs, instead of being synchronized, show a slight shift and the *sauteur* rabbits never perform the jumping. This discoordination dramatically impairs efficient locomotion, and as a consequence, individuals from this strain adapt their locomotion behavior for longer and/or faster movements by lifting the hindlimbs off the ground and move supported solely by their forelimbs, similarly to a human acrobat when walking on hands (S1 Movie). Additional anatomical problems have also been described in *sauteur* rabbits, they are born blind because of retinal dysplasia and start developing cataracts after their first year of life (Fig 1B) [16,20]. The *sauteur* phenotype, including abnormal gait and ocular lesions, has a simple genetic basis controlled by a single autosomal recessive allele ($s^{am}$) [14,16].

Here, we investigated the genetic basis of the striking gait behavior of *sauteur* rabbits. The abnormal gait of this strain is explained by a mutation in a splice donor site in the RAR related orphan receptor B gene (*RORB*) that causes the incorporation of intronic sequence in several aberrant transcript isoforms. In *sauteur* rabbits, the number of RORB-positive spinal cord neurons is drastically reduced when compared to wild-type rabbits. Moreover, *sauteur* rabbits also showed defects in the differentiation of populations of spinal cord interneurons including Dmrt3 –expressing neurons that have a well-established role in controlling gait [9].

## Results

### Genetic mapping of the abnormal gait behavior in *sauteur* rabbits

We employed a bulked segregant analysis [21] to map the causal locus for the *sauteur* phenotype. This approach provides a simple way for identifying genomic regions underlying a

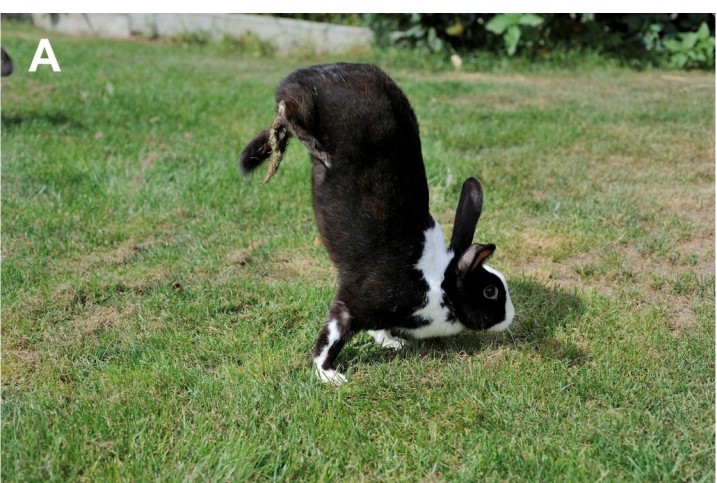
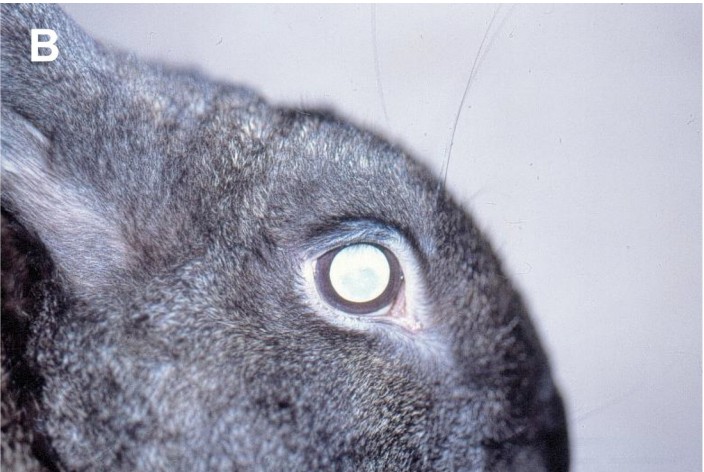

**Fig 1. The *sauteur d'alfort* strain and associated phenotypes. (A)** Typical posture of a *sauteur* rabbit ($s^{am}/s^{am}$) adopted when jumping (i.e., moving faster or across longer distances). Hindlegs are lifted from the ground, the body is held vertically, and locomotion is achieved through the alternate use of the forelegs. **(B)** Ocular malformations observed both in $s^{am}/s^{am}$ and $+/s^{am}$ individuals include bilateral papillary colobomas, reduction in pupillary reflexes, bilateral cataracts with lesions in various components of the eye, glaucoma, and/or entropion and ectropion. Photo credits: **(A)** R. Cavignaux; **(B)** S. Boucher.

phenotype of interest by pooling DNA samples generated from experimental crosses according to their phenotype. To this end, we crossed a single male of the *sauteur D'Alfort* strain, expected to be homozygous ($s^{am}/s^{am}$), with a single female of the New Zealand white breed homozygous for the wild-type allele (+/+). We produced an F2 population comprising 52 animals and the proportion of homozygous mutant (23%) did not deviate significantly from that expected for an autosomal recessive phenotype. Bulked DNA samples were created by pooling DNA of *sauteur* and non-*sauteur* individuals into two separate pools, followed by whole-genome sequencing (see S1 Table for details). Sequence reads were mapped to the rabbit reference genome sequence [22], resulting in an average effective coverage of 37.6X for the pool containing the individuals exhibiting the *sauteur* phenotype and 36.5X for the wild-type pool.

To screen the genome for regions of elevated genetic differentiation between the two pools (Fig 2A), as expected at the *sauteur* locus, we extracted read counts from the sequencing data and estimated allele frequency differentiation ($\Delta AF$) using a sliding-window approach. We averaged $\Delta AF$ along the genome in overlapping windows of 5,000 SNPs iterated every 1,000 SNPs, for a total number of 9,405 windows (median size = 1.01 Mb). The average allele frequency differentiation between pools across the genome was low ($\Delta AF = 0.13$) and a single region on chromosome 1 showed highly elevated levels of genetic differentiation. This region contained 94.7% of the windows (89 out of 94) in the top 1% of the empirical distribution of $\Delta AF$ ($\Delta AF \geq 0.41$) and encompassed a large segment of genome (~65Mb; chr1:25,520,137–91,295,391bp). The remaining five windows in the top 1% were located on two scaffolds (ChrUn0030 and Chr0044) that are currently unplaced in the assembly of the rabbit genome sequence and most likely located on chromosome 1 according to this result. Linkage and scaffolding information from ongoing efforts in our labs to improve the rabbit genome supports this notion.

As a complement to the genetic differentiation analysis, we estimated pooled heterozygosity ($H_p$) across the genome using an identical sliding window approach as described above. Since individuals exhibiting the *sauteur* phenotype ($s^{am}/s^{am}$) are expected to be identical-by-descent, and thus have low levels of heterozygosity nearby the causative locus, we calculated a ratio by dividing $H_p$ in the wild-type pool by $H_p$ in the *sauteur* pool. While heterozygosity across the genome was similar in both pools (0.43 vs. 0.42), as expected for groups of F2 animals, we

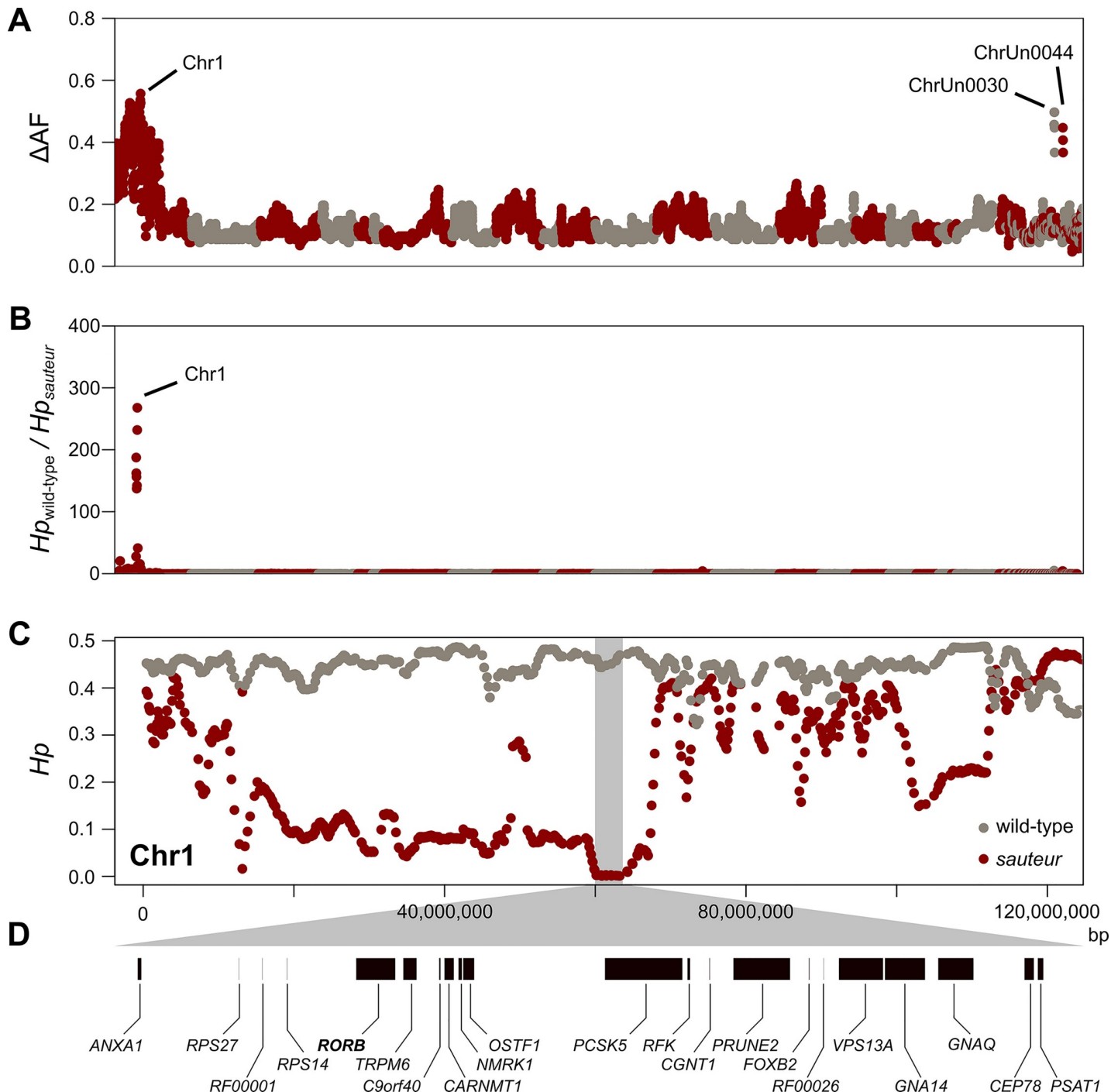

**Fig 2. Genetic mapping of the *sauteur* allele using experimental crosses and whole-genome analyses. (A)** Genetic differentiation (ΔAF) between the *sauteur* and wild-type pools across the genome. Each dot represents ΔAF averaged in windows of 5,000 SNPs with 1,000 SNPs steps. All scaffolds of the reference genome containing at least one valid window are presented along the x-axis in alternate colors. **(B)** Ratio of pooled heterozygosity ($H_p$) in the *sauteur* pool by $H_p$ in the wild-type pool across the genome. Each dot represents the ratio averaged in windows of 5,000 SNPs with 1,000 SNPs steps. All scaffolds of the reference genome containing at least one valid window are presented along the x-axis in alternate colors. **(C)** Close-up of $H_p$ across a large portion of chromosome 1 represented separately for the *sauteur* pool (red dots) and the wild-type pool (gray dots). The shaded area represents the candidate region where $H_p$ is extremely low in the *sauteur* pool. **(D)** Genes located within the candidate region.

observed highly elevated values of the estimated ratio in a region partially overlapping the region on chromosome 1 identified above (Fig 2B). The interval on chromosome 1 as defined by the windows in the top 1% of the empirical distribution of heterozygosity was again large (~56 Mb; chr1:11,880,652–67,488,194bp). However, the values were much more extreme in a region of 5.4 Mb (Chr1: 59,560,684–64,953,774bp), where we observed a >100-fold reduction of heterozygosity in the *sauteur* pool (Fig 2C). Furthermore, the *sauteur* group showed essentially no heterozygosity in this region as expected for the region harboring the causal mutation. This chromosomal interval contains 21 protein-coding genes (Fig 2D and S2 Table).

## A splice site mutation in *RORB* is associated with the *sauteur* phenotype

Using the whole-genome sequencing data, we next searched the 5.4 Mb candidate region for potential causative mutations, including small single-base changes and indels, as well as structural changes (inversions, copy number variation, and large indels). We specifically searched for variants (i) characterized by high $\Delta AF$ between the *sauteur* and wild-type pools–assuming a recessive mode of inheritance we expected a $\Delta AF = 0.75$ –and (ii) that were not present in whole-genome sequencing data of 14 populations samples of wild rabbits and six domestic breeds obtained as part of an earlier study [22].

Among candidate structural variants detected using several approaches (see Methods), our analysis revealed that either the variants were only weakly differentiated between the two pools based on the counts of split-reads, counts of abnormal read-pair orientation, and read depth, or were not considered *bona fide* structural rearrangements after careful examination. We also detected 69 point mutations and small indels with a potential impact on protein function (nonsynonymous, frameshift, stop gain, stop lost, and splice-site mutations), but 61 had a $\Delta AF \leq 0.5$ between the *sauteur* and the wild-type pools, and are therefore unlikely to explain the phenotype. Among the remaining 8, there was a splice site mutation in *RORB* with a $\Delta AF = 0.76$, which was the only mutation from the 69 candidates that was absent from other populations of wild and domestic rabbits. This variant corresponds to a change from GT to AT in the 5' donor site of intron 9 (chr1: 61,103,503bp; Fig 3A). A multiple sequence alignment showed that the splice mutation occurs in a genomic position that is completely conserved across 70 eutherian mammals (Fig 3B). The mutation may disrupt the normal splicing of *RORB*, a member of the NR1 subfamily of nuclear hormone receptors. *Rorb*-deficient mice suffer from retinal degeneration and exhibit motor impairments, characterized as a "duck-like" gait [12,23]. This gene is therefore an excellent candidate to explain the abnormal gait behavior and the presence of ocular lesions in *sauteur* rabbits.

Next, we genotyped the splice site mutation in rabbits from our experimental cross (12 *sauteur* and 40 wild-type) and in seven additional *sauteur* individuals obtained from two different breeders. This genotyping revealed that, in every case, individuals exhibiting the *sauteur* phenotype were homozygous for the splice mutation, whereas the individuals exhibiting the wild-type phenotype, with two exceptions, were either heterozygous (n = 26) or homozygous (n = 12) for the reference allele. The two discordant individuals from our cross, $s^{am}/s^{am}$ classified as having wild-type phenotype, can be explained by incomplete penetrance due to other interacting genetic factors, or by mis-phenotyping. The latter is the most likely explanation, since we phenotyped the individuals at a very young age (~4 weeks) when the abnormal gait in some individuals was still subtle and inconstant, which could lead to *sauteur* individuals being classified as wild-type.

## A high proportion of aberrant *RORB* transcripts in *sauteur* rabbits

To investigate the potential consequences of the candidate mutation in the splicing of *RORB*, we next amplified and sequenced *RORB* cDNA from spinal cord and retina of wild-type,

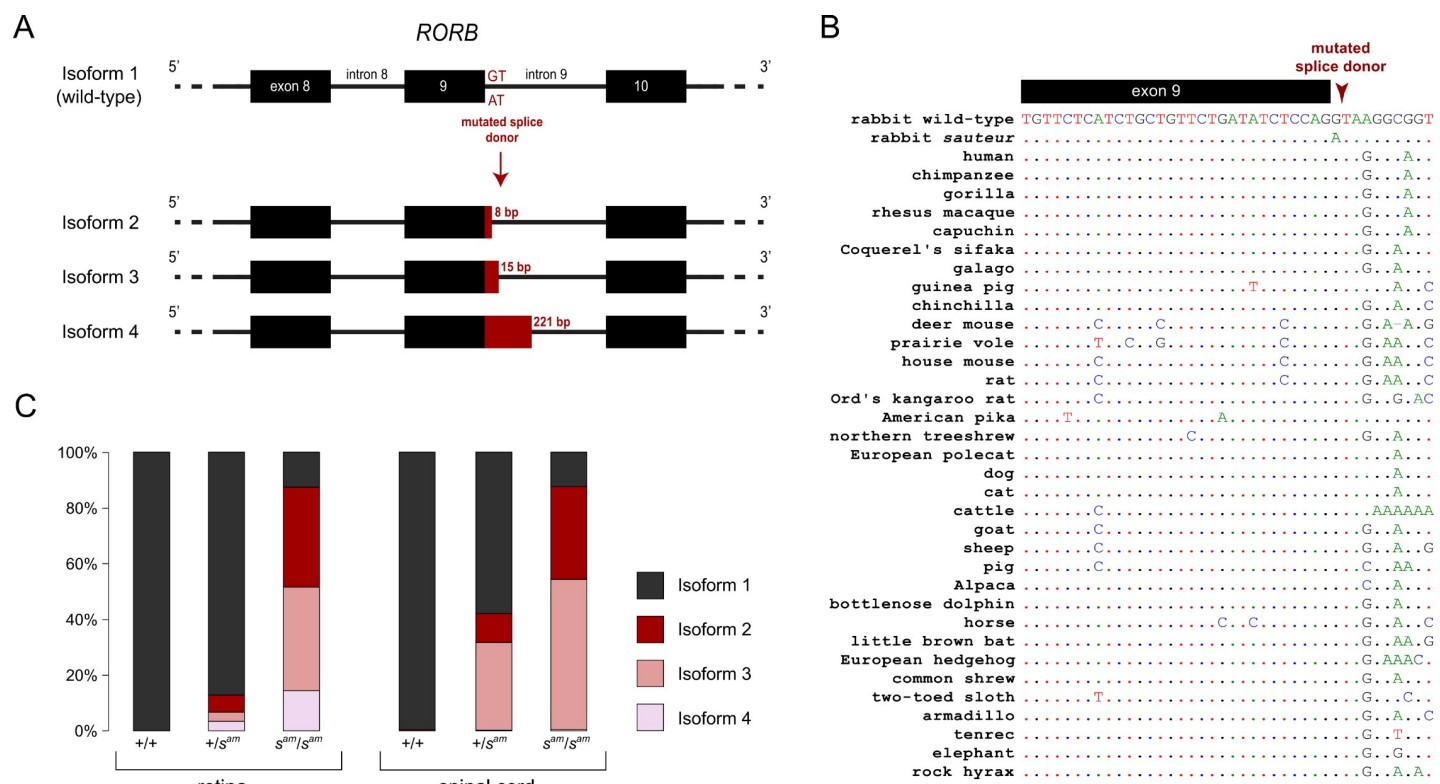

**Fig 3. A splice site mutation affects the expression of *RORB* in *sauteur* rabbits.** (**A**) The identified mutation in the splice donor site at the end of exon 9 results in three main mutant isoforms, which incorporate varying lengths of intronic sequence into the transcript. (**B**) Alignment of mammalian sequences at the causal locus, evidencing total conservation of the splice-site donor except for mutant *sauteur* rabbits. Only a subset of the 70 mammalian species analyzed are presented. (**C**) Relative abundance of the four main isoforms of *RORB* mRNA in the retina and spinal cord of rabbits of the three possible genotypes at the *sauteur* locus. Wild-type (+/+), heterozygous (+/$s^{am}$), and *sauteur* ($s^{am}$/$s^{am}$).

heterozygous, and *sauteur* rabbits using Nanopore technology (number of reads per individual and tissue ranged from 401 to 6,823; S3 Table). Among all samples and tissues, we recovered four isoforms, including the canonical *RORB* transcript and three alternatively spliced cDNAs incorporating intronic sequence (Fig 3A and S3 Table). The non-canonical transcripts seem to result from alternative GT splicing sites that lead to the incorporation of 8 (isoform 2), 15 (isoform 3), and 221 (isoform 4) nucleotides of intron 9. Isoforms 2 and 4 contain stop codons. In the wild-type individual, virtually all transcripts were identical to the canonical form (100% in the retina and 99.6% in the spinal cord). In contrast, the *sauteur* individual expressed the three non-canonical transcripts at high frequency in both tissues (>87.4%). The heterozygous individual had an overall splicing pattern intermediate between that observed in wild-type and *sauteur* individuals. The presence of a high proportion of aberrant transcripts in *sauteur* rabbits strongly suggests that the mutated splice site of *RORB* is causal.

The presence of transcripts carrying premature stop codons at a relatively high frequency could result in nonsense-mediated mRNA decay. If this occurs, the expression levels of *RORB* mRNA should be substantially lower in *sauteur* individuals. To test this, we quantified the expression of *RORB* mRNA in the retina and spinal cord of rabbits of all three genotypes using quantitative reverse transcription polymerase chain reaction (RT-qPCR). However, we found that the levels of *RORB* mRNA expression were similar among genotypes (S1 Fig).

## RORB-positive neurons are drastically reduced in number in the spinal cord of *sauteur* rabbits

To determine if and how the presence of RORB-positive neurons is affected in the *sauteur* rabbits, immunohistochemistry (IHC) was performed on the spinal cord of newborn rabbits from our experimental cross. Since the *sauteur* locomotor phenotype is not observable at birth, each individual was genotyped for the splice-site mutation of *RORB*. In rabbits homozygous for the wild-type allele, RORB is localized in the nucleus of a population of dorsal horn neurons ([Fig 4A]). These neurons are mainly situated in lamina III/IV, just below the Calbindin-expressing neurons of lamina II ([Fig 4B]). Moreover, around 40% of these neurons also co-expressed LBX1, a marker for the dI4 to dI6 spinal cord populations ([Fig 4B and 4C]) [24].

In the spinal cord from rabbits heterozygous for the *sauteur* allele (+/$s^{am}$), the number of neurons expressing RORB was approximately 25% lower than in the wild-type animals ([Fig 4D and 4E]). In contrast, in rabbits homozygous for the *sauteur* allele ($s^{am}$/ $s^{am}$), the expression of RORB was undetectable by IHC ([Fig 4D]). This suggests that the high proportion of abnormal transcripts in the spinal cord of *sauteur* rabbits results in a drastic reduction of RORB-positive neurons when compared to wild-type and heterozygous rabbits. This defect may cause the anomalous motor phenotype observed in *sauteur* rabbits.

## The differentiation of spinal cord interneuron populations is affected in *sauteur* rabbits

In mice, spinal RORB interneurons were shown to receive inputs from LTMRs (low threshold mechanoreceptors), which are primary sensory neurons localized in the dorsal root ganglia [25]. Moreover, in mice, RORB is involved in neuronal differentiation during development,

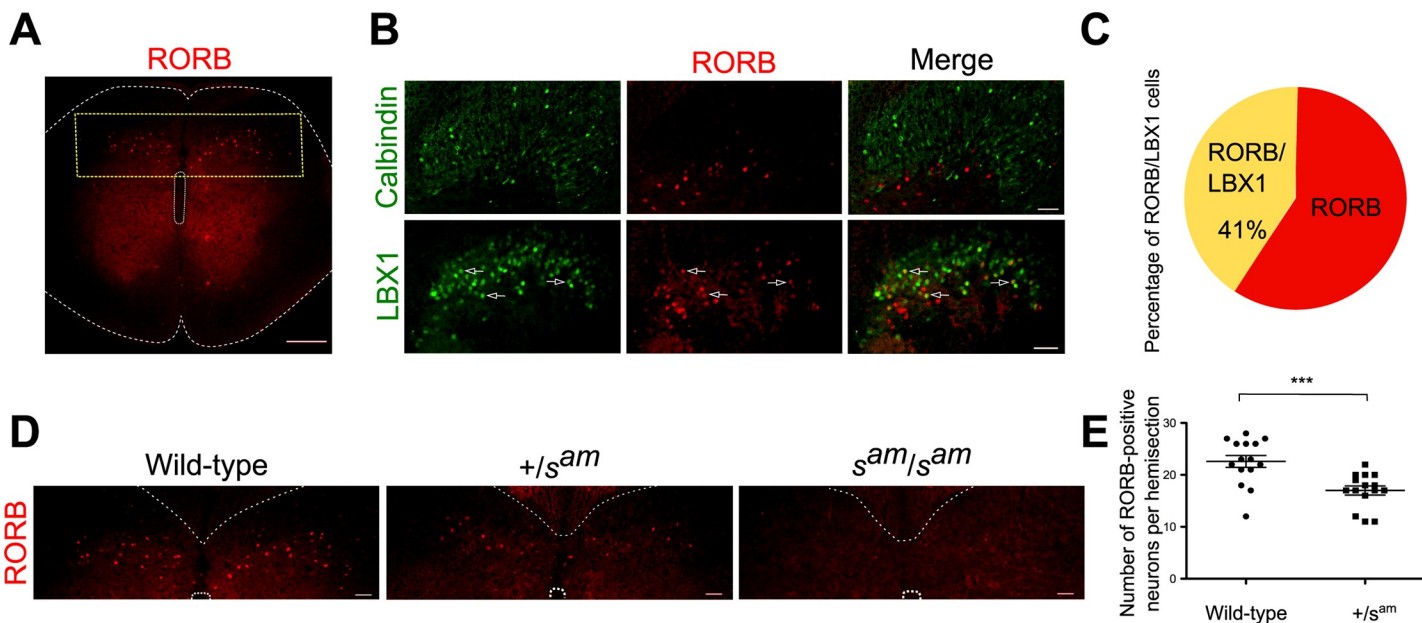

**Fig 4. The number of RORB-positive neurons are drastically reduced in the spinal cord of *sauteur* rabbits.** Immunohistochemistry (IHC) on newborns rabbit spinal cord. **(A)** In wild-type rabbits, RORB-immunopositive neurons are localized in the spinal cord dorsal horn (n = 5 animals, yellow dot rectangle = magnified areas in D). **(B and C)** Most of the RORB-immunopositive neurons are localized below Calbindin-expressing neurons and around 40% of them co-expressed LBX1 (arrows) (n = 547 cells from five wild-type animals). **(D and E)** In *sauteur* animals ($s^{am}$/$s^{am}$), the number of RORB immunopositive neurons was strongly decreased (no IHC staining, n = 6 animals) and in the heterozygous animals (+/$s^{am}$) the number of neurons is decreased by approximately 25% (n = 379 cells from 3 animals). (Two-tailed Mann Whitney test, $P$ = 0.0007) (Scale bars: 200μm for A and 50μm for B and D).

especially for the differentiation of photoreceptors and interneurons in the retina as well as the differentiation of the layer II/III and layer IV in the neocortex [26–28]. To determine if RORB plays a similar role in regulating cell differentiation in the rabbit spinal cord, we performed IHC to analyze different spinal cord neuronal populations. First, we investigated two interneuron populations of the dorsal horn that are localized close to RORB-expressing neurons and receive inputs from LTMRs and/or proprioceptive neurons. Calbindin is a marker for different interneuron populations in layer II and III of the dorsal horn where many interneurons receiving inputs from LTMRs are localized. In *sauteur* rabbits, the number of Calbindin-expressing neurons located in the layer V and VI appeared to be slightly larger in *sauteur* animals compared to wild-type. However, the number of cells expressing Calbindin in the rest of the spinal cord did not seem to be affected (Fig 5A and 5A').

In mice, SATB2-expressing interneurons are localized mainly in layer III to V and conditional mutant mice for SATB2 are characterized by a hyperflexion of the ankle joint during the early swing phase as well as a maintained flexion posture following sensory stimulations [29]. By using an antibody targeting SATB1 and SATB2 we observed a reduction of the number of SATB1 and/or SATB2-expressing interneurons in the dorsal horn layer I to III of the *sauteur* rabbits. The number of SATB1 and/or SATB2-expressing neurons in the other laminas was not affected (Fig 5A"). In contrary, the number and location of the motor neurons, labeled with an antibody against ChaT, was not altered in the *sauteur* rabbits (Fig 5B and 5B').

The locomotor phenotype of the *sauteur* rabbits is mainly occurring when the animals are moving at moderate to high speed. In mice, DMRT3-expressing interneurons, were shown to belong to the locomotor central pattern generators and contribute to hindlimb coordination during high-speed locomotion. In wild-type newborn rabbits, the DMRT3 interneurons are situated in the ventro-medial part of the spinal cord, a location similar to where these neurons are found in mice (corresponding to lamina VII and VIII in mice) (Fig 5C) [9]. The DMRT3 immunostaining was mainly found in the nuclei of the neurons except in some cells where it was found in the cytoplasm (Fig 5C). Moreover, they were localized at the level or below the central canal with some few exceptions (yellow arrowheads). In three out of six *sauteur* rabbits, many Dmrt3-expressing neurons were found outside their normal location above the central canal and in many of these neurons Dmrt3 was located in the cytoplasm rather than in the nucleus (Fig 5C and 5C'). Moreover, in those three animals the number of DMRT3-expressing neurons located at the level or below the central canal was also higher compared to wild-type animals (Fig 5C'). For the remaining three *sauteur* rabbits, the number of Dmrt3-expressing neurons was also higher than in the wild-type animals but they were normally located at the level or below the central canal and Dmrt3 was localized to the nucleus (Fig 5C and 5C'). These data indicate that RORB is involved in the differentiation of at least three populations of interneurons in the rabbit spinal cord.

## Discussion

In the present study, we show that a splice site mutation at the first nucleotide in intron 9 of the *RORB* transcription factor gene is causing the remarkable *sauteur* phenotype. Firstly, this was the only sequence variant identified by whole genome sequencing that fulfilled criteria for causality, including (i) an almost complete concordance with the *sauteur* phenotype among all samples tested (deviation from complete concordance is most likely due to misphenotyping), and (ii) the mutation is not found in previously reported sequences of a large number of wild and domestic rabbits [22]. Secondly, the mutated nucleotide position is completely conserved among all 70 eutherian mammals for which sequence information is available. Finally, a characterization of transcript isoforms by cDNA sequencing revealed that the presence of this

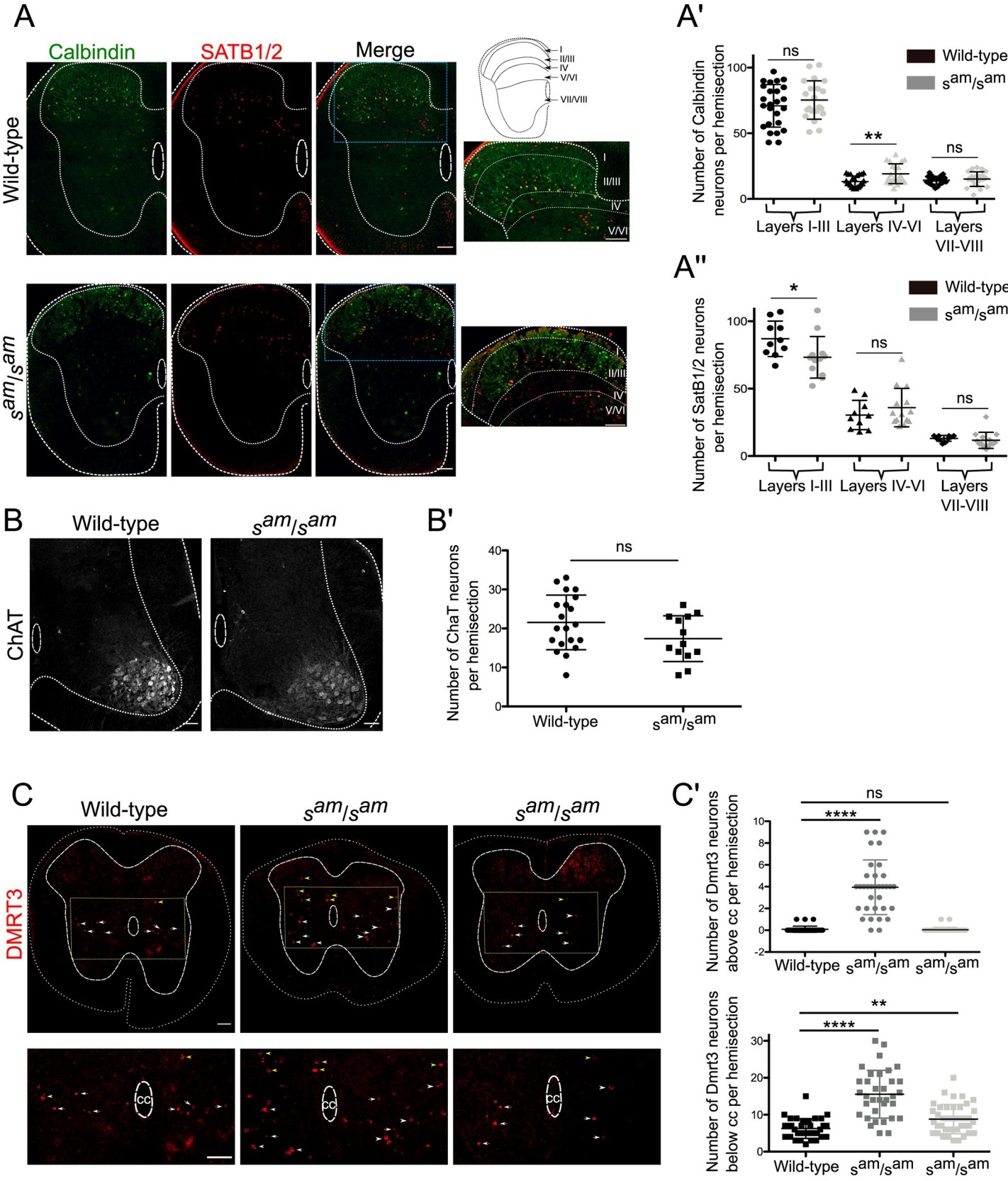

**Fig 5. In *sauteur* rabbits the differentiation of dorsal horn interneurons and DMRT3-expressing neurons is disturbed but motor neurons are not affected.**
Immunohistochemistry (IHC) on newborns rabbit spinal cord. **(A)** Localization of Calbindin and SATB1/2-expressing interneurons in wild-type and *sauteur* rabbits ($s^{am}/s^{am}$) spinal cords. The blue dotted rectangles in the merge images show the magnification of the dorsal horn depicted in the right panels. The spinal cord schematic displays the localization of the different laminas. **(A')** Quantification of the number of Calbindin-expressing neurons per hemisection in the different laminas of the spinal cord (n = 2 wild-type, 16 sections and n = 3 *sauteur*, 19 sections; two-tailed Mann Whitney test, $P = 0.49$ for layers I-III, $P = 0.005$ for layers IV-VI, and $P = 0.42$ for layers VII-VIII). **(A")** Quantification of the number of SATB1/2-expressing neurons per hemisection in the different laminas of the spinal cord (n = 2 wild-type, 9 sections and n = 2 *sauteur*, 14 sections; two-tailed Mann Whitney test, $P = 0.02$ for layers I-III, $P = 0.32$ for layers IV-VI, and $P = 0.13$ for layers VII-VIII). **(B)** Location of motor neurons in the lumbar spinal cord of wild-type and *sauteur* rabbits. **(B')** Quantification of the number of ChAT-expressing motor neurons per hemisection of spinal cord (n = 1 wild-type, 19 sections and n = 2 *sauteur*, 13 sections; two-tailed Mann Whitney test, $P = 0.09$). **(C)** Location of Dmrt3 neurons in wild-type rabbit (left panel) as well as in two sauteur rabbits (middle and right panels). **(C')** Quantification of Dmrt3-expressing neurons in wild-type rabbits (left column), in the three *sauteur* rabbits with misplaced Dmrt3 neurons (middle column correspond to the middle panel in Fig 5C) and the three other *sauteur* rabbits (right column correspond to the right panel in Fig 5C) (n = 2 wild-type, 21 sections, n = 2 *sauteur* for the middle column, 24 sections, and n = 2 *sauteur* for the right column, 22 sections). Two-tailed Mann Whitney test for the number of cells above the central canal $P<0.0001$ between wild-type and *sauteur* in middle column and $P = 0.23$ between wild-type and *sauteur* in right column. Two-tailed Mann Whitney test for the number of cells below the central canal $P<0.0001$ between wild-type and *sauteur* in middle column and $P = 0.004$ between wild-type and *sauteur* in right column. Scale bars for Fig 5A, 5B and 5C: 100µm. cc: central canal.

mutation is associated with aberrant splicing of the *RORB* gene and immunohistochemistry indicates a drastic reduction of RORB-positive neurons in the spinal cord of *sauteur* rabbits.

The aberrant isoforms 2 and 3 constitute 40% and 50% of the *RORB* transcripts present in the spinal cord of *sauteur* rabbits, respectively (Fig 3C). Isoform 2 is out of frame after exon 9 and is thus expected to result in a truncated RORB protein, while isoform 3 contains 15 extra nucleotides and is thus in frame and expected to result in a full-length protein with five extra amino acids inserted between the parts encoded by exon 9 and 10. Furthermore, RT-qPCR analysis using spinal cord from *sauteur* rabbits did not reveal any reduced level of *RORB* mRNA expression, indicating that the aberrant out-of-frame isoforms are not affected by non-sense mediated RNA decay. This also suggests that regulatory mutations altering the expression of the *sauteur* allele are unlikely to contribute to the phenotype.

In wild-type rabbit spinal cord, RORB-positive interneurons are localized in layer III/IV just below the Calbindin-expressing neurons and around 40% of them co-expressed LBX1. Moreover, some of them are situated more medial and just above the central canal suggesting that they belong to lamina V. This result is consistent with the localization of RORB interneurons in mice and rat, where they were shown to be situated in layer III/IV and layer V of the spinal cord and partially co-expressed with LBX1 [12,30,31]. In contrast, in *sauteur* rabbits we did not observe any RORB-expressing neurons, and in the heterozygous animals, the number of RORB-positive neurons was reduced by approximately 25%. By cDNA sequencing, we determined that in the spinal cords from heterozygous animals, the aberrant isoforms 2 and 3 represent roughly 40% of the total mRNA which can explain the reduced RORB protein expression in those animals. The antibody we have used recognizes the part of the protein encoded by exons 5 and 6. Even if this epitope is localized upstream of the splice mutation identified in the *sauteur* rabbits we cannot rule out the possibility that a truncated protein might still be expressed but undetectable with our antibody. It is therefore possible that the aberrant splice forms encode proteins that are not folded correctly and therefore degraded. However, an alternative explanation is that the presence of defect RORB proteins results in a loss of RORB-positive neurons which causes the *sauteur* phenotype.

The causality of the *RORB* splice site mutation in *sauteur* rabbits is further supported by the phenotypic overlap with *Rorb* knock-out mice, which show retinal degeneration and a duck-like gait [12,23]. Further dissection of the $Rorb^{-/-}$ phenotype in mice showed that the gait phenotype is replicated by selective inactivation of RORB-positive inhibitory interneurons and that these are required for a fluid walking gait [12]. In mice, the spinal cord RORB-expressing interneurons were shown to be part of the LTMR-RZ (low threshold mechanoreceptor recipient zone), a region involved in receiving inputs from Aβ, Aγ and C-LTMR primary sensory neurons and transmitting innocuous touch perception, such as texture discrimination and

hairy skin tactile sensitivity [25]. Different population of interneurons that belong to this LTMR-RZ region were shown to be involved in the regulation of fine motor control. Indeed, it was demonstrated that in mice the ablation of the spinal cord RORα or Zic2 interneurons impaired their ability to walk on a thin beam [32,33]. Moreover, conditional mutant mice deficient for SATB2 have a slight hyperflexion phenotype of the ankle joint at the beginning of the swing phase and a maintained hyperflexion of the limbs after Von Frey or Hargreaves stimulation [29]. In mice, Rorb was shown to play an essential role in cell differentiation in particular in the cortex and in the retina. By consequence, *RORB* could play a similar function in cell differentiation in the spinal cord. Calbindin is a marker for several populations of interneurons located in layer II and III that receive a variety of inputs from LTMRs [34]. In *sauteur* rabbits, there is a slight increase in the number of Calbindin-expressing interneurons in laminas IV to VI. SATB2-expressing interneurons are located close to RORB interneurons and are also part of the LTMR-RZ. Our IHC analysis demonstrated that there is a moderate decrease in the number of SATB1 and SATB2-expressing interneurons in the spinal cord of *sauteur* rabbits compared to control animals. Thus, the altered differentiation of the LTMR-RZ region is a possible cause of the excessive hindlimb lifting in *sauteur* rabbits.

In *Rorb*$^{-/-}$ mice, the hindlimb hyperflexion phenotype does not seem to be accompanied with a loss of the left-right alternation pattern whereas, at high-speed velocity, the hindlimbs of the *sauteur* rabbits are desynchronized during the swing phase. Indeed, a kinematic study showed that instead of synchronizing hindlimb movements when performing a hop, the hindlimbs of the *sauteur* rabbits showed a slight desynchronization when lifted from the ground, which suggest a potential alteration in the locomotor central pattern generator [19].

DMRT3-expressing spinal interneurons are known to play an essential role in regulating coordination of the hindlimb movements in horses and mice [9,35]. *Dmrt3* null mutant mice switch between left-right alternation and synchronous movements of their hindlimbs when they run at high speed on a treadmill. In the spinal cord of *sauteur* rabbits, the number of neurons expressing DMRT3 is larger than in the controls, and in three out of six *sauteur* rabbits many of these neurons were misplaced above the central canal. In addition, in these three animals, the DMRT3 protein was localized to the cytoplasm of many neurons rather than in the nucleus. Altogether these data suggest that the *RORB* mutation causes differentiation defects of spinal cord interneuron populations involved in the transmission of mechanoreception and regulation of locomotion.

In the mouse spinal cord, the RORB-expressing interneurons play an important role in gating proprioceptive sensory information by ensuring the presynaptic inhibition of the primary afferents [12]. A similar mechanism taking place in the rabbit spinal cord and causing the locomotion phenotype observed in the *sauteur* rabbit is possible, but currently not known. In addition to its expression in the spinal cord, RORB is also expressed in many regions in the brain such as the primary somatosensory, auditory, visual and motor cortex, in some thalamus and hypothalamus nuclei, in the pituitary gland and in the superior colliculus [30]. Thus, we cannot exclude the possibility that an alteration of *RORB* function in the brain contributes to the locomotion phenotype characteristic for the *sauteur* rabbits.

Strong inter-individual variability in the locomotion phenotype has been observed among *sauteur* rabbits [19]. In some individuals, the aberrant phenotype can be quite weak, and despite the loss of synchrony of their hindlimbs, their swing phase is similar to that of other rabbits. Other individuals show a stronger phenotype, i.e., they lift their hindlimbs over the head and walk only on their forelimb. The incomplete penetrant phenotype we found regarding the misplaced DMRT3 neurons in three out of six *sauteur* rabbits might be related with the observed variability of the locomotion phenotype. In mice, DMRT3 expression is strongest during development and declines shortly after birth, whereas the locomotor phenotype of the

*sauteur* rabbit becomes evident when they are 1 to 2 months old. This precludes the possibility to explore the correlation between the strength of the locomotor phenotype and the misplaced DMRT3 neurons using immunohistochemistry.

In conclusion, this study demonstrates that a mutation in the *RORB* gene is the cause of the locomotion phenotype observed in *sauteur* rabbits, likely through aberrant differentiation of spinal interneurons.

## Methods

### Ethical statement

The experimental procedures were approved by the Ethical Committee for Animal Research of the University of Castilla la Mancha, Spain (Register number CEEA: 1012.02). Rabbits were kept under standard conditions of housing with unrestricted access to food and water, according to the European Union Directive no. 86/609/CEE.

### Experimental crosses

The parental generation consisted of a cross between a *sauteur* male ($s^{am}/s^{am}$) and a wild-type female belonging to the New Zealand white breed (+/+). We produced six F1 individuals (three males and three females; $+/s^{am}$), which were crossed with each other to generate an F2 generation. The cross resulted in 40 individuals exhibiting the wild-type phenotype (+/+ or $+/s^{am}$) and 12 individuals exhibiting the *sauteur* phenotype ($s^{am}/s^{am}$). The distribution of phenotypes did not deviate significantly from the one expected for an autosomal recessive mutation. The F2 individuals were phenotyped between 3–4 weeks of age after weaning. Each individual was placed isolated in a cage and its movement was observed for five minutes and classified as *sauteur* or wild-type. No genotype information was available at this point, so classification was blind to this.

### Whole genome sequencing

Genomic DNA was isolated from blood or ear punches using an EasySpin Genomic DNA Tissue Kit SP-DT-250 (Citomed, Lisbon, Portugal), and RNA was removed with a RNAse A digestion step. Two DNA pools (*sauteur* and wild-type) were generated by pooling equimolar amounts of DNA of the different individuals. These two bulks were then used to generate paired-end sequencing libraries using the TruSeq DNA PCR-free Library Preparation Kit (Illumina, San Diego, CA) according to manufacturer's protocols. The resulting libraries were sequenced on an Illumina HiSeq X instrument using 2x150 bp reads. Whole-genome sequencing data are available in the Sequence Read Archive (www.ncbi.nlm.nih.gov/sra) under the bioproject PRJNA559371.

### Read mapping and variant calling

After sequencing, read quality was inspected with *FastQC* v0.11.8 [36]. To remove Illumina adapters and low-quality sequences, we used *Trimmomatic* v0.38 [37] with the following parameters: TRAILING (used to remove low quality bases from the 3' prime end): 15; SLIDING WINDOW (trims a read when the average quality within a window is below a defined threshold): 20–4; MINLEN (removes reads shorter than a minimum length): 30. After removing adapter and low-quality reads, the trimmed reads were further rechecked for quality using *FastQC* and then mapped to the rabbit reference genome assembly (OryCun2.0) using *BWA--MEM* v0.7.17-r1188 [38] with default settings. Sequence alignment files were filtered for unpaired reads and checked for quality of mapping and coverage using *SAMtools* [39] and custom scripts.

Variant calling was carried out using a Bayesian haplotype-based method as implemented in *Freebayes* v1.2.0 [40]. The ploidy parameter was set to 24, which is twice the number of individuals in the *sauteur* pool. However, for the wild-type pool, given the large number of individuals incorporated, to reduce the computational burden the ploidy variable was set to 40. We modified the following additional parameters relative to the default settings: minimum mapping quality of 40, minimum base quality of 20, and a minimum coverage of 10X. To avoid calling variants overlapping repetitive elements or mis-assembled segments of genome, positions with a read count two times higher than the average coverage across the genome were discarded. Allele counts for subsequent analysis were extracted for each variant.

## Genetic mapping using genetic differentiation and heterozygosity statistics

To identify the genomic region containing the *sauteur* allele, we took a two-folded approach based on different aspects of the data. The following analyses were restricted to biallelic SNPs with a quality score of 200 or greater as estimated by *Freebayes*, which resulted in a total of 10,534,832 markers. First, we estimated genetic differentiation for each SNP across the genome using the absolute difference in allele frequency between pools ($\Delta AF$). The values for individual SNPs were then averaged across the genome in overlapping windows of 5,000 SNPs iterated every 1,000 SNPs. Windows with less than 4,000 SNPs, which occurred at the end of scaffolds, or in small scaffolds containing fewer SNPs, were excluded from the analysis. Second, we estimated genetic diversity for the same windows using the pooled heterozygosity ($H_P$) statistic as described by [41]. The statistic was calculated for each pool independently and then transformed into a ratio by dividing $H_p$ in rabbits exhibiting the wild-type phenotype by $H_p$ in rabbits exhibiting the *sauteur* phenotype.

## SNP annotation and structural rearrangements

The annotation of the detected variants (both SNPs and indels) was performed using the genetic variant annotation and effect prediction toolbox *SnpEff* [42]. We screened the genome for variants with moderate or high impact as predicted by *SnpEff*, which includes nonsynonymous, frameshift, stop gain, stop lost, and splice-site mutations. In addition, we screened our candidate genomic interval for structural variants, including deletions, insertions, duplications, inversions and translocations, using three complementary approaches that explore different aspects of the sequencing read data: 1) *Breakdancer* [43], which uses read pair orientation and insert size; 2) *DELLY*, which uses paired-end information and split-read alignments [44]; and 3) *LUMPY* [45], which uses a combination of multiple signals including paired-end alignment, split-read alignment, and read-depth information. All candidate structural variants reported in the candidate interval were visually inspected in the Integrative Genomics Viewer (*IGV*) (v2.4.10) [46].

## SNP genotyping

To genotype the candidate splice-site mutation, we amplified a small amplicon followed by Sanger sequencing. Primer sequences are given in S4 Table. In addition to the individuals obtained from our cross, we sampled and genotyped seven *sauteur* individuals from two different breeders. Genomic DNA extractions were performed as described above for whole-genome sequencing.

## Isoform analysis using Nanopore sequencing

We investigated alternative splicing of *RORB* in three adult rabbits of the three possible genotypes (homozygous for the *sauteur* allele [$s^{am}/s^{am}$], heterozygous [$+/s^{am}$], and homozygous for the wild-

type allele [+/+]). Rabbits were deeply sedated with a mixture of xylacin (Rompun, 8 mg/kg; Bayer) and ketamine (Imalgene 1000, 40 mg/kg; Merial) administered intramuscularly. Euthanasia was performed with an intracardiac injection of thiopental (Thiopental 0.5 g, 100 mg/kg; B. Braun) as previously described [47,48]. From these individuals, we extracted the spinal cord and retina. Total RNA was isolated from both tissues and purified using the RNeasy Mini Kit (QIAGEN). We performed an extra RNase-Free DNase digestion step to remove any contaminating DNA, followed by estimation of RNA concentration and purity using Qubit RNA BR assay kit. After RNA isolation, cDNA was generated by reverse transcribing ~1 μg of RNA using the GRS cDNA Synthesis Kit (GRiSP, Porto, Portugal) following the manufacturer's protocols.

To identify potential splicing differences of *RORB* between *sauteur* and wild-type rabbits, we designed primers that spanned the annotated transcript from the rabbit reference genome from exon 7 to 11 (S4 Table). These primers were 5'-tailed to allow for individual barcoding through a two-step PCR approach based on [49]. The first PCR reaction was prepared with approximately 25 ng DNA, 5 μL 2x Qiagen MasterMix, 0.4 μL of 10 pM of each primer and 3.2 μL PCR-grade water, and was run under the following conditions: 1) an initial denaturing step of 95˚C for 15 min; 2) 5 touch-down cycles with 95˚C denaturing for 30 s, a 64–60˚C annealing temperature touch-down for 30 s and 72˚C extension temperature for 45 s; 3) 35 cycles with 95˚C denaturing for 30s, a 60˚C annealing step for 30 s and 72˚C extension for 45 s; 4) a final extension at 60˚C during 20 min. We set up the second (barcoding) PCR reaction using 2 μL of PCR product, 5 μL 2x Qiagen MasterMix, 1 μL of a mix of individually labeled primers with P5/P7 binding sites and 1 μL of PCR-grade water. The following program was used for the barcoding PCR: 1) an initial denaturing step of 95˚C for 15 min; 2) 10 cycles with 95˚C denaturing for 5 s, a 55˚C annealing temperature step for 20 s and a 72˚C extension for 45 s; 3) a final extension at 60˚C during 20 min. Each PCR product was cleaned using AMPure XP beads (0.7:1 bead-to-sample volume ratio), DNA was quantified, and all samples were pooled at equimolar concentrations for sequencing. The sequencing library was prepared using the Ligation Sequencing Kit (SQK-LSK109) following the manufacturer's protocol for short amplicons. The library was run for two hours on a MinION 9.4.1 flow cell (Oxford Nanopore).

To filter out any sequenced non-target DNA, we started by mapping all reads to a custom reference consisting of the transcript sequence using *MINIMAP2* [50], a general purpose aligner that is suited for mapping reads with high error rates and that is also splice-aware (see below). With this approach we identified 58,538 individual sequences mapping to the *RORB* transcript. Since we did not use standard MinION barcodes, we demultiplexed the samples by re-converting the reads that mapped to the transcript and retaining only those that had an unaltered full 7-bp barcode sequence plus an additional 10-bp of the adapter overhang (to account for the high sequencing error rate of Nanopore sequencing). Reads were then remapped using *MINIMAP2* to a new reference sequence containing the full *RORB* open reading frame. We inspected sequencing reads from each sample using *IGV* (v2.4.10) [46] and detected four main transcript isoforms resulting from changed splice site locations between exons 9–10 (see Results section). The relative abundance of each transcript was obtained by analyzing sequencing coverage using SAMtools *mpileup* function, considering the counts of each transcript in positions where two or more transcripts share sequence. Demultiplexed fastq reads from each individual/tissue have been deposited in GenBank under the bioproject PRJNA559371.

## RT-qPCR

To assess levels of *RORB* mRNA expression across the three genotypes, we used quantitative reverse transcription polymerase chain reaction (RT-qPCR). As described above, we sampled retina and spinal cord tissue from one individual of each genotype (+/+, +/$s^{am}$, $s^{am}$/$s^{am}$),

extracted total RNA and reverse transcribed it to cDNA. We designed PCR primers to amplify two separate amplicons, each amplicon spanning exon-exon boundaries. One amplicon spanned exons 7–8 (211 bp), and the other spanned exons 10–11 (146 bp). Results for each amplicon were normalized to the expression of a housekeeping gene (*GAPDH*) using a -ΔCq approach [51]. Three replicate assays were performed for each amplicon/tissue/individual combination. A list of the primers used for qPCR can be found in S4 Table.

### Immunostaining

We performed immunostaining on newborn rabbits (24–48 hours after birth) obtained from a cross between two carriers of the *sauteur* allele (+/$s^{am}$). Individuals were euthanized as described above. Using a peristaltic pump (TPU2AD; Aalborg) we then flushed the blood out of the vascular system using a PBS solution and tissues were fixed applying a cardiovascular perfusion using a 10% solution of neutral buffered formalin. After perfusion, heads, internal organs and esophagus were extracted. In similar way, skin, ribs and muscle were removed to access the vertebral column. A transverse cut was made along the entire vertebral column avoiding damage to the spinal cord, which was extracted after cutting the roots and connective tissues. Next, we post-fixed the tissue in paraformaldehyde 4% 24h at 4˚C, and after 24h of fixation, we washed the tissue at least three times for 15 min with PBS solution. The fixed tissue was stored at 4˚C until sectioning. To select individuals from all genotypes for subsequent experiments, each individual was genotyped for the splice-site mutation in *RORB*. We performed experiments on five individuals homozygous for the wild-type allele, six heterozygotes (+/$s^{am}$) and six animals homozygous for the *sauteur* allele ($s^{am}$/$s^{am}$).

Next, part of the tissue was cryoprotected in a gradient of sucrose (10%, 20% and 30%), frozen in cryomedium (Killik, Bio-Optica) and 20 μm sections were performed on a cryostat (Cryocut 1800, Leica). For IHC, the tissue was washed in PBS (1X) before incubation in blocking buffer 5% donkey serum, 3% BSA in PBS (1X) 0.3% Triton for 1h at room temperature. The primary antibodies were incubated in the blocking buffer overnight at 4˚C. The primary antibodies are mouse anti-RORB 1/200 (R&D systems PP-N7927-00), GP anti-LBX1 1/1000 (gift from Carmen Birchmeier), mouse anti-SATB1/2 1/250 (Abcam ab51502), rabbit anti-Calbindin 1/1000 (Swant CB38), goat anti-ChAT 1/100 (Merck AB144P) and GP anti-DMRT3 1/1000 as in [9]. After PBS (1X) washing, the secondary antibodies were also incubated in the blocking buffer for 1h at room temperature. The secondary antibodies were all used at a 1/1000 dilution: Goat anti-Guinea Pig Alexa fluor 594 (Invitrogen A11076), Donkey anti-Mouse Alexa fluor 488 (Invitrogen A21202), Donkey anti-Mouse Alexa fluor 594 (Invitrogen), Donkey anti-Goat Alexa fluor 647 (Invitrogen) and Donkey anti-Rabbit Alexa fluor 488 (Invitrogen A21206). After washing in PBS (1X), the tissue was mounted using Prolong Diamond Antifade (ThermoFisher Scientific P36961). Pictures were acquired using the OlympusBX61WI fluorescent microscope with the Volocity software (Quorum Technologies). Image analysis was performed with ImageJ, Adobe photoshop CC and figures were made using InkScape.

All statistics were performed using GraphPad Prism software and two tailed Mann-Whitney tests. Significance symbols used are * *P*-value<0.05, ** *P*-value<0.01,*** *P*-value < 0.001 and **** *P*-value<0.0001.

### Supporting information

**S1 Fig. Gene expression levels of RORB in the retina and spinal cord.** Gene expression was measured through quantitative RT-qPCR of two amplicons in three rabbit individuals, one per genotype: wild-type (+/+), heterozygote (+/$s^{am}$) and *sauteur* ($s^{am}$/$s^{am}$). The *y*-axes indicate a relative measure of expression of each amplicon controlled for the expression of a housekeeping gene

(*GAPDH*). Main bars indicate average relative expression, and error bars indicate the minimum and maximum values of three technical replicates for each tissue/individual.
(PDF)

**S1 Table. Whole genome resequencing and read mapping statistics.**
(PDF)

**S2 Table. List of genes within the candidate region (chromosome1:59,560,684–64,953,774 bp).**
(PDF)

**S3 Table. Relative abundance of the four most common *RORB* isoforms quantified through Nanopore sequencing of amplicons obtained from cDNA of retina and spinal cord.** Each cell indicates the percentage of reads of each isoform (read counts are shown in parenthesis).
(PDF)

**S4 Table. List of primers used in this study.**
(PDF)

**S1 Movie. Patterns of locomotion in sauteur rabbits.** From Samuel Boucher.
(MP4)

## Acknowledgments

We thank Bernardino Silva for help with animal breeding.

## Author Contributions

**Conceptualization:** Miguel Carneiro, Samuel Boucher, Nuno Ferrand, Klas Kullander, Leif Andersson.

**Formal analysis:** Miguel Carneiro, Jennifer Vieillard, Pedro Andrade, Sandra Afonso, João Branco.

**Funding acquisition:** Miguel Carneiro, Leif Andersson.

**Investigation:** Miguel Carneiro, Jennifer Vieillard, Pedro Andrade, Samuel Boucher, Sandra Afonso, José A. Blanco-Aguiar, Nuno Santos, João Branco, Pedro J. Esteves, Nuno Ferrand, Klas Kullander, Leif Andersson.

**Methodology:** Miguel Carneiro, Jennifer Vieillard, Pedro Andrade, Sandra Afonso, José A. Blanco-Aguiar, Nuno Santos.

**Project administration:** Miguel Carneiro, Leif Andersson.

**Resources:** Miguel Carneiro, Samuel Boucher, José A. Blanco-Aguiar, Pedro J. Esteves, Nuno Ferrand, Klas Kullander, Leif Andersson.

**Supervision:** Miguel Carneiro, Klas Kullander, Leif Andersson.

**Validation:** Jennifer Vieillard.

**Visualization:** Miguel Carneiro, Jennifer Vieillard, Pedro Andrade, Samuel Boucher.

**Writing – original draft:** Miguel Carneiro, Jennifer Vieillard, Klas Kullander, Leif Andersson.

**Writing – review & editing:** Miguel Carneiro, Jennifer Vieillard, Klas Kullander, Leif Andersson.

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
