## [Decision Letter · Decision Letter 0]

22 Nov 2020

Dear Dr Carneiro,

Thank you very much for submitting your Research Article entitled 'Retinoid-related orphan nuclear receptor RORB is required for saltatorial locomotion in rabbits' to PLOS Genetics. Your manuscript was fully evaluated at the editorial level and by two independent peer reviewers. The reviewers appreciated the attention to an important problem, but raised some substantial concerns about the current manuscript. Based on the reviews, we will not be able to accept this version of the manuscript, but we would be willing to review a revised version. We cannot, of course, promise publication at that time.

In particular, it will be important to fully address each of the comments by Reviewer #2.

If you decide to revise the manuscript for further consideration at PLOS Genetics, please aim to resubmit within the next 60 days, unless it will take extra time to address the concerns of the reviewers, in which case we would appreciate an expected resubmission date by email to plosgenetics@plos.org.

[LINK]

We are sorry that we cannot be more positive about your manuscript at this stage. Please do not hesitate to contact us if you have any concerns or questions.

Yours sincerely,

Gregory P. Copenhaver

Editor-in-Chief

PLOS Genetics

Gregory Barsh

Editor-in-Chief

PLOS Genetics

Reviewer's Responses to Questions

**Comments to the Authors:**

Reviewer #1: The manuscript describes a very interesting study in a very peculiar rabbit line/strain, the sauteur d'Alfort rabbit, that exibits a defective locomotion behaviour.

The authors provided several evidences supporting the causative role of a splice site mutation in the RORB gene opening new lines of research toward the complete characterization and role of RORB in gait.

The authors identified the putative causative mutation by whole genome resequencing by applying a bulked segregant analysis based on an F2 family and reported gene expression and histochemical evidences.

I have just a couple of suggestions:

1) The title could be misleading - I would suggest to readdress it to what the work was able to demonstrate - that means that a mutation affecting the RORB gene expression determines an anomalous locomotion.

2) Sauter d'Alfort rabbits are traditionally considered as a strain or as a line and not a breed. This might be corrected in the introduction and in the abstract.

Reviewer #2: In this study by Carneiro et al. use a domestic rabbit breed, the sauteur d’Alfort, which has an abnormal bipedal locomotor gait. Whole genome sequencing analysis identified a mutation at a single locus containing RORB, which could explain this abnormal gait. I am not in the position to critically evaluate the validity of the genetic analysis and sequencing and therefore I will focus my critique on the IHC data and their interpretation. This study is of interest and potentially suitable for publication in PLOS Genetics, there are however some weak points that need to be addressed in a revised manuscript.

1) Figure 4: The changes in the number of neurons is mostly limited to the dorsal part of the spinal cord. Most of the neurons involved in the generation and coordination of locomotion are located in the ventral part of the spinal cord. Is there any explanation as to whether this is the case? Some discussion of these observations is necessary.

2) The results of Figure 5 are not convincing and need to be quantified appropriately. High magnification pictures with quantification should be provided for all panels to allow the reader to assess these results. I suggest quantifying the number of neurons along the dorso-ventral axis.

3) DMRT3 expression and distribution was only affected in 3 of 6 rabbits while the remaining 3 were not affected. This raises doubts about the validity of these results and whether the observed changes in DMRT3 expression contributes to the phenotype. The conclusion that the differentiation of DMRT3 neurons is affected in these animals is not fully supported.

**Have all data underlying the figures and results presented in the manuscript been provided?**

Reviewer #1: Yes

Reviewer #2: **No: **

PLOS authors have the option to publish the peer review history of their article (what does this mean?). If published, this will include your full peer review and any attached files.

Reviewer #1: No

Reviewer #2: No

---

## [Decision Letter · Decision Letter 1]

17 Feb 2021

Dear Dr Carneiro,

We are pleased to inform you that your manuscript entitled "A loss-of-function mutation in RORB disrupts saltatorial locomotion in rabbits" has been editorially accepted for publication in PLOS Genetics. Congratulations!

Yours sincerely,

Gregory P. Copenhaver

Editor-in-Chief

PLOS Genetics

Gregory Barsh

Editor-in-Chief

PLOS Genetics

Comments from the reviewers (if applicable):

Reviewer's Responses to Questions

**Comments to the Authors:**

Reviewer #2: The authors have addressed all my concerns and comments.

**Have all data underlying the figures and results presented in the manuscript been provided?**

Reviewer #2: Yes

PLOS authors have the option to publish the peer review history of their article (what does this mean?). If published, this will include your full peer review and any attached files.

Reviewer #2: No

**Data Deposition**

http://datadryad.org/submit?journalID=pgenetics&manu=PGENETICS-D-20-01498R1

**Press Queries**

---

## [Editor Report · Acceptance letter]

28 Feb 2021

PGENETICS-D-20-01498R1 

A loss-of-function mutation in RORB disrupts saltatorial locomotion in rabbits 

Dear Dr Carneiro, 

We are pleased to inform you that your manuscript entitled "A loss-of-function mutation in RORB disrupts saltatorial locomotion in rabbits" has been formally accepted for publication in PLOS Genetics! Your manuscript is now with our production department and you will be notified of the publication date in due course.

With kind regards,

Alice Ellingham

PLOS Genetics

On behalf of:
